# Study of the Effects of Condensed Tannin Additives on the Health and Growth Performance of Early-Weaned Piglets

**DOI:** 10.3390/ani14162337

**Published:** 2024-08-14

**Authors:** Min Ma, Yuriko Enomoto, Tomotsugu Takahashi, Kazuyuki Uchida, James K. Chambers, Yuki Goda, Daisuke Yamanaka, Shin-Ichiro Takahashi, Masayoshi Kuwahara, Junyou Li

**Affiliations:** 1Animal Resource Science Center, Graduate School of Agricultural and Life Sciences, The University of Tokyo, Kasama 319-0206, Japan; aminma@g.ecc.u-tokyo.ac.jp (M.M.); aenoyan@g.ecc.u-tokyo.ac.jp (Y.E.); tomtaka@g.ecc.u-tokyo.ac.jp (T.T.); 2Veterinary Pathophysiology and Animal Health, Graduate School of Agriculture and Life Science, The University of Tokyo, Tokyo 113-8654, Japan; akuwam@g.ecc.u-tokyo.ac.jp; 3Laboratory of Veterinary Pathology, Graduate School of Agriculture and Life Science, The University of Tokyo, Tokyo 113-8654, Japan; auchidak@mail.ecc.u-tokyo.ac.jp (K.U.); achamber@mail.ecc.u-tokyo.ac.jp (J.K.C.); 4Laboratory of Cell Regulation, Graduate School of Agriculture and Life Science, The University of Tokyo, Tokyo 113-8654, Japan; leonardo.fibonacci11235813@docomo.ne.jp (Y.G.); atkshin@mail.ecc.u-tokyo.ac.jp (S.-I.T.); 5Laboratory of Food and Physiological Models, Graduate School of Agriculture and Life Science, The University of Tokyo, Kasama 113-8654, Japan; adyama@mail.ecc.u-tokyo.ac.jp

**Keywords:** quebracho tannin, early-weaned piglets, growth performance, diarrhea, free amino acids

## Abstract

**Simple Summary:**

Tannins, astringent polyphenols found in plants, show potential as a natural antibiotic to act as a substitute for antibiotics in animal feed additives. For ruminants, it has been widely documented that the addition of tannins to feed improves feed efficiency by increasing the amount of bypass protein. However, for monogastric animals, tannins are widely recognized as antinutritional factors, and, in some regions, it is still common practice to remove tannins from feed, e.g., through silage. The results of our previous study showed that the addition of 0.2% and 0.3% MGM-P (MGM is a commercial brand of tannin), especially the 0.3% addition, provided preventative effects regarding the incidence of diarrhea in early-weaned piglets. It has also been shown to have the ability to improve villus morphology and alleviate piglet diarrhea. Therefore, this study evaluated the effectiveness of higher doses of tannin extract MGM-P (0.5% and 1.0%) in preventing diarrhea and improving the growth performance of weaned piglets. Comparisons were also made with antibiotic additives. The results suggest that an addition level of 0.5% shows potential as an alternative to the use of antibiotics in monogastric animal feed.

**Abstract:**

Using 0.5% and 1.0% MGM-P, the objective of the present study was to determine a more appropriate additive level for early-weaned piglets as an alternative to the use of antibiotics. Thirty-six weaned piglets were allotted to one of four groups and given a basal diet (NC), with the basal diet containing either 0.5% (LT) or 1.0% (HT) MGM-P or antibiotics (PC). Diarrhea incidence, growth performance, hematology, blood biochemistry, and blood amino acid concentrations were monitored during the experimental period. Three piglets per group with a body weight nearest to the average level were slaughtered after the experiment to assess their organ index. The results showed that no diarrhea was observed either in the treatment groups or in the control group. The 0.5% group showed an upward trend in body weight and average daily gain at all stages. The WBC counts at 21 days of age were higher (*p* > 0.05) both in the MGM-P addition groups and the LT and HT groups. For some of the plasma amino acids, such as arginine, phenylalanine concentrations were significantly lower (*p* < 0.05) in the HT group at the end of the trial. The pathological examination of all organs confirmed no differences. Consequently, the 0.5% MGM-P addition level may be suggested as a potential alternative to the use of antibiotic additives. Even with additives as high as 1%, there is no negative effect on ADG and FCR.

## 1. Introduction

The early weaning of piglets is used to increase the reproductive efficiency of sows by maximizing the number of sows that deliver and increasing their slaughter weight each year. However, this process can result in post-weaning diarrhea (PWD), growth retardation, and even the death of early-weaned piglets, which brings huge economic losses to the pig industry [1]. Considering efficacy and cost, since the early 1950s, the use of antibiotics in the swine industry has been the most common practice worldwide [2]. Flavomycin is considered a relatively safe antibiotic because of its non-absorbability in the gastro-intestinal tract of animals and is widely used as a growth promoter in pig feed in most countries [3]. Wahlstrom et al. supplemented weaned piglets with 2 mg/kg of flavomycin in the diet and found an upward trend in average daily weight gain during both the growing and fattening periods and that high doses of the antibiotic resulted in faster growth during the fattening period [4]. However, drug-resistant microorganisms resulting from the excessive use of antibiotic additives over time have debilitated the curative effectiveness of clinically important antibiotics in human and animal medicine, threatening human health [5].

Over the past several decades, various alternatives to antibiotics and additional measures have been tried to reduce the use of antibiotics [6]. Despite the wide variety of projects under investigation, few alternatives can completely replace antibiotics in practice without posing any risk. Tannins are naturally occurring astringent polyphenols in plants with antimicrobial properties. The properties of tannins, as a natural antimicrobial, are attributed to their ability to combine with extracellular microbial enzymes to inhibit their activity [7]. This process has neither a specific target, nor do tannins have access to the inside of the cell; therefore, it is relatively difficult for this process to cause drug resistance. Nevertheless, this characteristic also renders tannins susceptible to exhibiting anti-nutritional properties through their binding to feed proteins and digestive enzymes [8]. Furthermore, they also have anti-oxidative [9] and anti-inflammatory [10] properties, which may help improve intestinal barrier function in piglets.

MGM-P is one of the condensed tannin (CT) products extracted from the heartwood of the quebracho tree (*Schinopsis lorentzii*). Compared to hydrolyzed tannins, condensed tannins are more structurally stable, which permits them to perform functions more permanently in the complex environment of the gastrointestinal tract. Su et al. [11] studied the effects of adding quebracho tannin to the diet of nursing pigs and found that the addition of tannin at the 0.1% level had no positive effect on the diarrhea incidence and growth performance of pigs. The results of our previous study showed that the addition of 0.2% and 0.3% MGM-P, especially the high addition level of 0.3% MGM-P, improved villus morphology and alleviated piglet diarrhea incidence [1]. Therefore, higher doses of MGM-P supplementation may have the potential to replace antibiotic additives. However, it should be noted that tannins can dose-dependently inhibit the utilization of feed amino acids in monogastric animals to produce antinutritional effects [12]. 

The aim of the present study was to evaluate higher doses of MGM-P (0.5% and 1.0%) in preventing the effects of diarrhea and improving the growth performance of weaned piglets. In our study, diarrhea incidence, growth performance, hematology, blood biochemistry, blood amino acid concentrations, and organ weights were measured.

## 2. Materials and Methods

The experiment described herein was conducted at the Animal Resource Science Center of the University of Tokyo (Kasama, Japan) and approved (P20-097) by the Animal Care and Use Committee of the Graduate School of Agricultural and Life Sciences at the University of Tokyo.

### 2.1. Materials

#### 2.1.1. Tannin

The condensed tannins in the quebracho tannin extract, MGM-P, were more than 50% (Table 1). The product was purchased from Kawamura Co., Ltd., Tokyo, Japan.

#### 2.1.2. Diet

The basal diet was purchased from KIMURA NOSAN SHOJI CO., LTD, Tokyo, Japan. The feed complied with National Research Council standards [13]. The ingredients (Table 2) and chemical composition (Table 3) were the same as those used in our previous publication [1].

#### 2.1.3. Animals and Experimental Design

Four gestating specific-pathogen-free sows were purchased from Nakamura Chikusan (Ibaraki, Japan) to obtain piglets (Duroc × Landrace × Yorkshire) for this study. All piglets were measured by birth weight and numbered. The male piglets were castrated at 14 days of age and all piglets were introduced to the basal diet to acclimatize. At 21 days of age, the 36 piglets were weaned and divided into four groups according to body weight and sex using the Experimental Animal Allotment Program in accordance with the method established by Kim and Lindemann [14] (Table 4). The negative control (NC) group received only the basal diet; the low-dose treatment (LT) and high-dose treatment (HT) groups received the basal diet with 5 g/kg and 10 g/kg MGM-P, respectively. The positive control (PC) group received the basal diet with 0.1 g/kg flavomycin^80^ (Huvepharma Japan Inc., Kyoto, Japan). 

One pen fed three piglets, and each treatment comprised three pens. The piglets in each pen were close in body weight. Each pen was equipped with a feed trough and a drinking bowl with a valve for ad libitum access to food and water. The experimental period was 21 days.

### 2.2. Methods

#### 2.2.1. Diarrhea Manifestations

Feces were observed twice daily at 9:00 a.m. and 3:00 p.m. The occurrence of diarrhea was determined when sloppy feces were found on two or more consecutive days.

#### 2.2.2. Growth Performance

The piglets were weighed and feed consumption was recorded at the same time before the experiment (d0) and on days 7, 14, and 21 of the experiment. Average daily feed intake (ADFI) and feed conversion rate (FCR) were calculated.

#### 2.2.3. Blood Sampling

Blood was collected from all 36 piglets from the jugular vein during weighing on days 0, 7, 14, and 21. A 21-gauge needle (VENOJECT II; Terumo, Tokyo, Japan) was used to harvest blood for storage in 5 mL collection tubes containing EDTA-Na.

#### 2.2.4. Blood Hematology Analysis

Hematology analyses, including white blood cell (WBC), lymphocyte, neutrophil, red blood cell, and platelet counts, were performed using a pocH-100iV Diff hematology analyzer (Sysmex Corp., Kobe, Japan).

#### 2.2.5. Plasma Collection and Biochemical Examination

After hematological analysis, the collected blood was centrifuged for 20 min (3000 rpm) at 4 °C to obtain plasma and then analyzed to determine biochemical parameters including glutamic pyruvic transaminase (GPT), glutamic oxaloacetic transaminase (GOT), gamma-glutamyl transferase (GGT), ammonia (NH_3_), blood urea nitrogen (BUN), amylase (AMYL), glucose (GLU), total protein (TP), and triglyceride (TG) using an automatic dry-chemistry analyzer (DRI-CHEM 3500s; Fujifilm, Tokyo, Japan).

#### 2.2.6. Plasma-Free Amino Acids

To prevent changes in the concentration of free amino acids in the collected blood, deproteinization was performed immediately after measuring blood biochemistry [15]. The blood was then stored at −80 °C until further analysis. In total, 20 amino acids were tested in the experiment, including 10 essential amino acids (arginine, histidine, isoleucine, leucine, lysine, methionine, phenylalanine, threonine, tryptophan, and valine), 3 semi-essential amino acids (cysteine, tyrosine, and glutamine), and 7 non-essential amino acids (aspartic acid, serine, alanine, glycine, glutamic acid, proline, and asparagine). The analysis was carried out using the LC/MS/MS Method Package for Primary Metabolites version 2.0 (Shimadzu, Kyoto, Japan) with a Shimadzu LCMS-8030 system. 

#### 2.2.7. Actual and Relative Weights/Lengths of Organs and Intestines

One medium-weight piglet per pen, with a total of three piglets in each treatment group, was selected after the feeding trial and sacrificed following the induction of deep anesthesia via thiopental sodium (Ravonal 0.5 g; Mitsubishi Tanabe Pharma, Osaka, Japan) injection into the jugular vein. Necropsies were performed, and the piglets’ organs (liver, pancreas, spleen, kidney, stomach, small intestine, and large intestine) were carefully removed. The weights of all organs and the lengths of the intestinal sections were measured. Relative organ weight was calculated as organ weight divided by BW (%), and the relative length of the intestinal section to piglet BW was also calculated (cm/kg).

#### 2.2.8. Statistical Analysis

Data analysis was performed using JMP Pro software (version 15.2.0, SAS Institute Inc., Cary, NC, USA). One-way analysis of variance was used to compare differences among the experimental groups. When the *p*-value from the analysis of variance was <0.05, pairwise differences were assessed using the Tukey–Kramer honestly significant difference test. The results of the experiment are presented as the means ± standard errors of the mean.

## 3. Results

### 3.1. Diarrhea

During the experimental period, no PWD was confirmed in piglets in all of the treatment groups, including the LT, HT, and PC groups and even the control group. Mushy feces were occasionally observed in the HT group, which was easily rectified in a short period of time, and no contagion was observed in the same pen.

### 3.2. Growth Performance

The final body weight changes are shown in Table 5. The piglets’ final average body weight in the control group was 17.68 kg; in comparison, in the LT group, this figure was 18.25; in the HT group, it was 17.35; and in the PC group, it was 17.77. The upward trend in body weight in the 0.5% addition group, however, was not significant. ADG also showed an upward trend in the LT group at 0.59; in comparison, in the NC group, it was 0.56; in the HT group, it was 0.55; and in the PC group, it was 0.57. The ADFI was 0.74 in the NC group, 0.74 in the LT group, 0.73 in the HT group, and 0.74 in the PC group. 

The weekly body weight changes are shown in Figure 1. During the experimental period, all piglets remained healthy, and no fatalities occurred. Although the changes in the LT group’s body weight showed an upward trend, this trend was not significant.

### 3.3. Blood Hematology Analysis

Information on the changes in blood hematology in the piglets after weaning can be observed in Table 6. There were no significant differences in blood hematology counts; however, a higher tendency of WBC and lymphocyte and neutrophil counts was observed in the CT additive groups and the LT and HT groups at the end of the experiment (*p* > 0.05).

### 3.4. Blood Biochemical Analysis

As shown in Table 7, no significant differences were observed in indicators related to liver metabolism, including GPT, GOT, GGT, and NH_3_ (*p* > 0.05). The BUN, AMYL, GLU, and TP concentrations were also not affected by the different treatments (*p* > 0.05). However, antibiotic supplementation significantly increased (*p* < 0.05) the TG concentration in the piglets’ plasma at 21 d in comparison with that of the NC and LT groups.

### 3.5. Plasma-Free Amino Acids

Information on the changes in the plasma-free amino acid concentrations of the weaned pigs is shown in Table 8. Dietary 1.0% MGM-P supplementation significantly reduced (*p* < 0.05) the concentration of arginine in the piglets’ blood at age 21 d compared with that of the PC group. In addition, piglets in the HT group had a significantly reduced (*p* < 0.05) phenylalanine concentration at age 21 d compared with piglets provided with a basal diet and a diet containing antibiotics. 

### 3.6. Actual and Relative Weights/Lengths of Organs and Intestines

No abnormalities were found in the piglets’ organs during necropsies performed at the end of the experiment. Information on the effect of dietary MGM-P supplementation on the relative weight or length of the organs and intestines of the piglets is presented in Table 9. The different dietary treatments had no influence on these relevant parameters of the piglet organs under examination (*p* > 0.05). 

## 4. Discussion

In the present study, no PWD was confirmed in piglets subjected to all of the treatments, including the LT, HT, and PC groups and even the control group. Yi et al. [16] reported that dietary 0.1% CT from kenwood supplementation decreased the diarrhea rate after the day of weaning to 28 days (*p* < 0.05), with no significant effect on growth performance. Su et al. [11] studied the effects of adding quebracho tannin to the feed of nursing pigs and found that the addition of tannin at the 0.1% level had no positive effect on the diarrhea incidence and growth performance of pigs. The results of the author’s previous study also proved that the addition of quebracho CT reduced the incidence of diarrhea, and the results also indicated that the diarrhea reduction effect was dose-dependent and 0.3% more efficacious than that reported for the 0.2% supplement [1]. This is one reason why an experiment involving a greater addition of tannin was conducted in the present study. 

One explanation for the lack of PWD being confirmed in piglets from all of the treatments could be that, in the present study, piglets were exposed and acclimatized to the solid form of the basal diet starting 7 days before weaning. This process may mitigate the stress induced by the piglets’ conversion of nutrients from breast milk to solid feed. Indeed, this process mitigated piglet feed intake and digestibility caused by the nutritional effect on the intestinal mucosa villus. Secondly, the basal diet used in the present study was commercial feed, containing several anti-PWD ingredients, such as probiotics, several other types of herbal extract, and a small proportion of high-level zinc sulfate, but without any antibiotics. The final zinc content was 119.6 mg/kg. Thus, to verify the real tannin effect on post-weaning diarrhea incidence and intestinal microorganisms in early-weaned piglets, we started an additional experiment involving a basal diet without probiotics and other herbal extract content in which the zinc level was consistent with the NRC level. Another reason for this finding is the fact that the present study was conducted in a university facility, where better conditions came from the lower rearing density and clean sanitary conditions, and these conditions helped to reduce the weaning stress to which the piglets were exposed. Prescott J.F. et al. [17] indicated in their study that antibiotics can only exert their greatest effect when the animals to which they are administered are in poor health and their living conditions are unhygienic. During the experiment, some piglets occasionally showed sticky and mushy feces in the HT group, which was easily rectified within a short period of time. Yi et al. [16] suggested in their study that a 0.1% CT addition could reduce the incidence of diarrhea to 16.7%. In the present study, sticky and mushy feces were observed in the HT (1% = 10 g CT/kg) group, although not frequently. In addition, no contagion was found in the same pen. This above phenomenon could be considered the result of indigestion. Thus, it is not known whether the occurrence of soft stools is associated with high levels of MGM-P, and the exact cause requires further confirmation. 

In the present study, we found that LT treatment showed an upward trend in ADG and ADFI with CT supplementation, but HT treatment showed a downward trend in ADG and ADFI, even if this figure was not significant. Tannin has been considered an anti-nutritional factor for a long period of time, especially CT. In their study, Ortizd et al. [18] fed chicks with feed containing 8 g/kg and 16 g/kg of faba bean tannin extract, which comprises condensed tannin, and found that it significantly affected the chicks’ growth performance, with 24-day weight gains of only 68% and 58% that of the normal diet group, respectively. Yi et al. [16] showed in their study that when administrating an additive of 0.1% condensed tannin, there was no significant effect on BW, ADG, ADFI, and F/G (*p* > 0.05). E. Seoni et al. [19] reported in their study that sainfoin, which contains a non-negligible amount of condensed tannin, is a suitable homegrown protein source for grower–finisher pigs and can be included at a rate of up to 15% to replace 7% soybean in a diet, without having any noteworthy effects on growth. Therefore, the CT additive level is critical in determining whether it produces antinutritional effects.

CT in several forage plants (e.g., L. *corniculatus* and *sulla*) has been shown to offer advantages for ruminants and result in increased milk production, wool growth, ovulation rate, and lambing percentage, as well as reduced bloat risk and reduced internal parasite burdens. When CT-bonded protein as a bypass protein enters the abomasum, the protein will be released and digested. Jones and Mangan [20] reported in their study that CT can bind with protein at near-neutral pH (pH 3.5–7.5) to form CT–protein complexes, which dissociate and release protein at a pH less than 3.5. Thus, in monogastric animals, whose stomach pH is usually less than 3.5, the appropriate CT additive could not increase the number of CT-bonded protein complexes high enough to affect protein digestibility. 

In the present study, 0.5% MGM-P addition resulted in an upward trend in ADG and ADFI compared to the antibiotic additive group. As mentioned above, the present study was conducted in a university facility where better conditions helped to reduce the incidence of several stresses to which the piglets were exposed during weaning, meaning that antibiotics could not exert their greatest effect. 

The RBC of the piglet blood obtained in our study, as with our previous study, was at a similar level to that reported by Czech et al. [21]. These results indicate that RBC is stable when CT is present. WBCs play a primary role in both fighting inflammation and clearing extracellular pathogens [22,23]. The onset of PWD is often accompanied by an increase in the number of WBCs in the blood of piglets [1,24]. In the present study, the WBC count in all of the treatments showed low levels on the weaning day, the age of 21 days, compared to those reported by Czech et al. [21]. These results prove that because all of the piglets used in the present study were in good health at the weaning stage, the white blood cell’s primary role in the body’s defense could not be shown. The values of the CT addition groups, both the LT and HT groups, gradually increased to a range of normal values consistent with our previous research results. However, the WBC count in the NC and PC groups showed a downward trend after weaning. The above results somehow suggest that tannin treatment affects WBC levels, and, thus, this effect requires further examination. 

Due to the relative lack of studies on the administration of CT additives to monogastric animals, in the present study, a relatively high additive level of 1% was used in order to determine the liver cell injury parameters of GOT, GPT, and GGT. The results in this regard did not show any anomalies, thus indicating that a supplement level as high as 1.0% is acceptable for animals. 

No differences in the piglets’ blood ammonia and urea nitrogen concentrations were confirmed. This result indicates that protein digestion and the functions of the liver and kidneys were not affected by CT addition. Ye et al. [25] found in their study that the addition of 50 mg/kg of flavomycin to the feed of Hy-Line Brown chickens resulted in elevated plasma triglyceride levels. Similar results were observed in the PC group at 21 d; however, levels in the chickens included in this particular group were not affected by CT addition. 

Regarding the plasma amino acid concentration at the end of the experiment, at the age of 21 days after weaning, the HT group showed significantly lower arginine and phenylalanine levels. Mariscal-Landín et al. [12] evaluated the effect of different tannin levels on the coefficient of apparent ileal digestibility of sorghum amino acids in growing pigs and found that tannin levels of up to 1.05% did not affect the digestion of arginine, whereas tannin levels of 4% or more inhibited its digestion. Arginine is one of the factors linked to growth hormone release in young children through the somatotropic axis and, if deficient, may affect early-stage growth [26]. In another study, when broiler diets were supplemented with 0.5%, 1.5%, 2%, and 2.5% mimosa tannins, 2.5% supplementation caused a significant decrease in the ileal digestibility of phenylalanine compared to the basal diet [27]. Phenylalanine is also necessary for the sufficient growth of weaned piglets [28,29]. Consequently, the weaker body weight gain observed in the HT group may have been related to the lower concentration of these two amino acids in the piglets.

In their study, Wang et al. [30] added 0, 0.05, 0.1, and 0.15% tannic acid and antibiotics to the diets of 21-day-old weaned piglets and found that neither tannic acid nor antibiotics had any effect on the piglets’ relative organ weight. The above results are consistent with those of our previous study [1] and the present work. Thus far, few studies have been conducted on the effect of high-level condensed tannin addition on the relative organ weight of piglets. The present research results show that the addition of 1.0% MGM-P still has no effect on the development of organs, and the pathological features of organs were not observed during dissection.

To summarize our study, the effectiveness of antibiotic additives is diminished under current feeding conditions. Supplementation of 0.5% MGM-P in piglet feed is expected to replace antibiotics. In the central role of tannins as antimicrobials as an alternative to antibiotics, it is necessary to investigate the effect of tannins on the intestinal microflora of piglets in future research.

## 5. Conclusions

In conclusion, according to the results of present research on MGM-P supplementation, there is a tendency to increase the nADG and FCR of piglets when the additive level is 0.5%, especially without antinutritional effects and anemia. Even with additives as high as 1%, there is no negative effect on ADG and FCR. The results on growth and health imply that the use of 0.5% MGM-P in early-weaned piglet diets has the potential to replace antibiotic additives.

## Figures and Tables

**Figure 1 animals-14-02337-f001:**
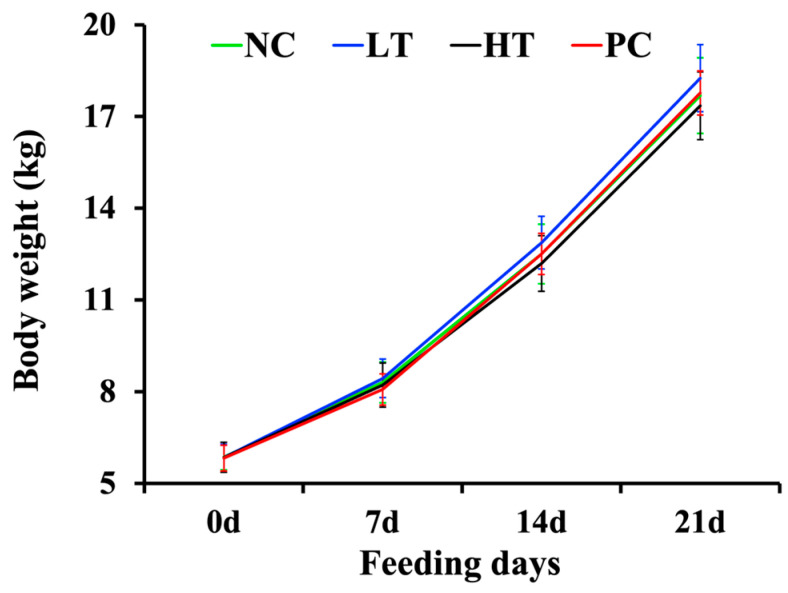
Effects of MGM-P supplementation on the body weight of the weaned piglets. Values are expressed as the mean ± SEM; *n* = 9. There were no statistically significant differences among the four groups based on the results of the one-way analysis of variance.

**Table 1 animals-14-02337-t001:** Technical specification of MGM-P.

Characteristic	Criterion
Polyphenols (mg catechin/g)	>500
Humidity (maximum)	15

**Table 2 animals-14-02337-t002:** Ingredients of basal diet (as-fed basis) ^1^.

Ingredient	Content (%)
Corn	34.45
Defatted milk powder	18.00
Fatty powder	6.20
Sugar	10.00
Soybean meal	25.00
Fish meal	4.50
Calcium diphosphate	0.20
Calcium carbonate	0.65
Salt	0.20
B vitamins	0.15
Vitamins A, D, and E	0.10
Trace minerals	0.15
L-lysine hydrochloride	0.06
DL-methionine	0.09
L-threonine	0.03
Copper sulphate	0.21
Vitamin K3	0.01
Total	100

^1^ The other diets were based on this diet.

**Table 3 animals-14-02337-t003:** Chemical composition of basal diet.

Chemical Composition	Content (%)	Amino Acid	Content (%)
DM	90.50	Contained	
CP	22.60	Arginine	1.32
EE	6.60	Histidine	0.63
CF	1.10	Isoleucine	0.99
Ash	5.60	Leucine	2.03
NFE	54.60	Lysin	1.56
DE (Mcal/kg)	3.70	Methionine + cysteine	0.83
Ca	0.81	Phenylalanine + tyrosine	1.92
NpP	0.45	Threonine	0.96
Na	0.26	Tryptophan	0.28
Cl	0.36	Valine	1.15
K	0.99	Digestible	
Mg	0.14	Arginine	1.22
Fe (mg/kg)	182.18	Histidine	0.58
Zn (mg/kg)	105.32	Isoleucine	0.88
Mn (mg/kg)	87.51	Leucine	1.83
Cu (mg/kg)	125.29	Lysin	1.42
I (mg/kg)	1.95	Methionine + cysteine	0.74
Se (mg/kg)	0.30	Phenylalanine + tyrosine	1.47
Vitamin A (IU/kg)	100,051.62	Threonine	0.85
Vitamin D (IU/kg)	2000	Tryptophan	0.25
Vitamin E (IU/kg)	20.04	Valine	1.01
Vitamin K (IU/kg)	0.57		
Thiamine (mg/kg)	5.15		
Riboflavin (mg/kg)	15.38		
Pantothenic acid (mg/kg)	27.83		
Nicotinic acid (mg/kg)	25.63		
Vitamin B6 (mg/kg)	5.93		
Choline (mg/kg)	1204.8		
Vitamin B12 (μg/kg)	21.88		
Biotin (mg/kg)	0.16		
Folic acid (mg/kg)	0.36		

Abbreviations: DM—dry matter; CP—crude protein; EE—ether extract; CF—crude fiber; NFE—nitrogen-free extract; DE—digestible energy.

**Table 4 animals-14-02337-t004:** Experimental animal allotment.

Group	NC	LT	HT	PC
Body weight	5.86	5.84	5.85	5.82
Number of piglets	9/(3pens)	9/(3pens)	9/(3pens)	9/(3pens)
SEM	0.42	0.44	0.49	0.42
*p*-value	1.00			
CV (%)	0.31			

Abbreviations: n—number of piglets; SEM—standard error of the mean; CV—coefficient of variation.

**Table 5 animals-14-02337-t005:** Effects of MGM-P supplementation on the growth performance of weaned piglets.

Measurement	NC	LT	HT	PC	*p*-Value
Initial weight (kg)	5.86 ± 0.42	5.84 ± 0.44	5.85 ± 0.49	5.82 ± 0.42	1.00
Final weight (kg)	17.68 ± 1.24	18.25 ± 1.1	17.35 ± 1.11	17.77 ± 0.72	0.97
ADG (kg/d)	0.56 ± 0.04	0.59 ± 0.03	0.55 ± 0.03	0.57 ± 0.02	0.80
ADFI (kg/d)	0.74 ± 0.05	0.74 ± 0.04	0.73 ± 0.06	0.74 ± 0.05	1.00
FCR (kg/kg)	1.31 ± 0.01	1.26 ± 0.02	1.33 ± 0.04	1.29 ± 0.03	0.34

Abbreviations: ADG—average daily gain; ADFI—average daily feed intake; FCR—feed conversion ratio. Data on the piglets’ weight and ADG are expressed as the mean ± SEM (*n* = 9); data on ADFI and FCR are expressed as the mean ± SEM (*n* = 3). There were no statistically significant differences among the four groups based on the results of the one-way analysis of variance.

**Table 6 animals-14-02337-t006:** Effects of MGM-P supplementation on blood hematology parameters in the weaned piglets.

Measurement		NC	LT	HT	PC	*p*-Value
0 d ^1^						
	WBC (×10^2^/μL)	81.8 ± 8.0	80.4 ± 7.3	75.6 ± 8.7	80.8 ± 6.9	0.95
	Lymphocyte (×10^2^/μL)	73.6 ± 7.7	70.6 ± 6.4	66.1 ± 7.4	73.0 ± 6.5	0.88
	Neutrophil (×10^2^/μL)	4.1 ± 0.4	4.6 ± 0.9	4.9 ± 1.1	3.6 ± 0.3	0.65
	RBC (×10^4^/μL)	482.0 ± 38.7	500.9 ± 22.5	549.8 ± 32.1	493.7 ± 18.3	0.38
	PLT (×10^4^/μL)	136.7 ± 20.8	148.6 ± 16.3	135.2 ± 20.0	158.7 ± 15.7	0.78
21 d						
	WBC (×10^2^/μL)	101.1 ± 8.9	127.9 ± 15.4	124.4 ± 16.2	107.9 ± 9.2	0.40
	Lymphocyte (×10^2^/μL)	69.7 ± 4.3	91.3 ± 9.2	87.1 ± 9.2	80.3 ± 7.8	0.22
	Neutrophil (×10^2^/μL)	19.0 ± 4.1	21.3 ± 4.3	22.6 ± 4.5	16.6 ± 0.9	0.71
	RBC (×10^4^/μL)	678.2 ± 16.1	704.4 ± 13.2	728.0 ± 9.8	709.8 ± 11.0	0.07
	PLT (×10^4^/μL)	76.0 ± 6.8	81.9 ± 2.7	71.2 ± 5.7	76.3 ± 4.8	0.56

Abbreviations: WBC—white blood cells; RBC—red blood cells; PLT—platelets. All data are expressed as the mean ± SEM; *n* = 9. ^1^ Blood was collected before the provision of feed with MGM-P on the day of weaning. There were no statistically significant differences among the four groups based on the results of the one-way analysis of variance.

**Table 7 animals-14-02337-t007:** Effects of MGM-P supplementation on the blood biochemical parameters of the weaned piglets.

Measurement		NC	LT	HT	PC	*p*-Value
0 d ^1^						
	GPT (U/L)	35.6 ± 1.1	31.4 ± 1.6	34.6 ± 1.1	33.4 ± 1.3	0.14
	GOT (U/L)	31.9 ± 2.4	32.3 ± 2.1	39.8 ± 7.4	32.6 ± 2.0	0.49
	GGT (U/L)	24.6 ± 4.6	22.7 ± 2.2	20.7 ± 1.6	21.7 ± 2.1	0.80
	NH_3_ (μg/dL)	121.0 ± 14.7	113.8 ± 4.7	150.2 ± 26.8	168.6 ± 40.3	0.40
	BUN (mg/dL)	6.8 ± 0.8	5.6 ± 0.3	6.0 ± 0.7	6.7 ± 0.6	0.47
	AMYL (U/L)	868.2 ± 102.2	971.2 ± 132.8	1060.2 ± 159.6	1040.7 ± 66.6	0.67
	GLU (mg/dL)	123.6 ± 2.6	123.2 ± 3.8	121.0 ± 4.0	132.7 ± 5.0	0.18
	TP (g/dL)	4.5 ± 0.2	4.8 ± 0.1	4.5 ± 0.1	4.9 ± 0.1	0.07
	TG (mg/dL)	53.4 ± 6.0	63.1 ± 13.7	47.8 ± 7.8	77.7 ± 15.4	0.29
21 d						
	GPT (U/L)	39.6 ± 1.3	36.0 ± 1.3	38.4 ± 2.2	38.3 ± 1.9	0.53
	GOT (U/L)	39.4 ± 3.7	38.6 ± 4.3	50.8 ± 13.6	51.1 ± 4.4	0.50
	GGT (U/L)	30.6 ± 3.0	32.3 ± 1.7	30.2 ± 1.5	30.4 ± 2.6	0.91
	NH_3_ (μg/dL)	110.4 ± 10.4	131.6 ± 11.8	151.2 ± 32.6	153.4 ± 14.7	0.38
	BUN (mg/dL)	16.4 ± 1.4	16.8 ± 1.2	14.7 ± 1.1	17.1 ± 0.8	0.47
	AMYL (U/L)	744.2 ± 82.7	868.0 ± 116.0	1013.9 ± 124.3	824.7 ± 42.8	0.27
	GLU (mg/dL)	141.3 ± 3.2	153.7 ± 8.8	148.2 ± 11.2	143.8 ± 4.1	0.68
	TP (g/dL)	5.0 ± 0.1	5.1 ± 0.1	5.0 ± 0.1	5.1 ± 0.1	0.52
	TG (mg/dL)	16.0 ± 1.3 ^b^	16.2 ± 1.7 ^b^	21.8 ± 3.6 ^ab^	35.2 ± 8.8 ^a^	0.03

Abbreviations: GPT—glutamic pyruvic transaminase; GOT—glutamic oxaloacetic transaminase; GGT—gamma-glutamyl transferase; NH_3_—ammonia; BUN—blood urea nitrogen; AMYL—amylase; GLU—glucose; TP—total protein; TG—triglyceride. All data are expressed as the mean ± SEM; *n* = 9. ^1^ Blood was collected before the provision of feed with MGM-P on the day of weaning. ^a, b^ Mean values within a row with dissimilar superscript letters are significantly different (*p* < 0.05).

**Table 8 animals-14-02337-t008:** Effects of MGM-P supplementation on plasma-free amino acid concentrations in the weaned piglets (uM).

Measurement	NC	LT	HT	PC	*p*-Value
0 d ^1^					
Asparagine	99.1 ± 7.3	85.0 ± 7.1	93.2 ± 5.3	99.9 ± 6.3	0.36
Aspartic acid	82.8 ± 2.5	80.9 ± 3.2	89.7 ± 2.6	83.4 ± 2.5	0.13
Serine	198.1 ± 7.7	176.7 ± 11.7	195.4 ± 10.6	215.2 ± 12.6	0.12
Alanine	542.4 ± 24.4	478.9 ± 39.1	502.4 ± 19.5	550.5 ± 32.1	0.29
Glycine	854.2 ± 47.3	804.0 ± 47.9	787.6 ± 35.3	854.4 ± 47.0	0.62
Glutamine	382.9 ± 20.7	336.2 ± 26.3	378.3 ± 13.0	363.8 ± 18.5	0.37
Threonine	241.8 ± 17.6	212.7 ± 18.3	234.3 ± 14.7	252.8 ± 18.0	0.42
Cysteine	3.5 ± 0.4	2.5 ± 0.5	3.6 ± 0.5	3.0 ± 0.6	0.39
Glutamic acid	102.3 ± 6.8	99.0 ± 9.4	106.2 ± 7.6	94.9 ± 6.2	0.75
Proline	264.4 ± 17.7	225.7 ± 21.2	234.0 ± 11.3	259.2 ± 21.8	0.39
Lysine	146.2 ± 11.0	117.3 ± 13.9	130.7 ± 8.6	144.9 ± 10.0	0.23
Histidine	35.3 ± 1.4 ^ab^	29.0 ± 2.9 ^b^	32.7 ± 1.5 ^ab^	37.5 ± 1.9 ^a^	0.04
Arginine	117.0 ± 8.5	91.6 ± 7.4	106.7 ± 7.0	111.2 ± 10.2	0.19
Valine	205.9 ± 13.3	171.3 ± 17.3	185.6 ± 13.1	208.7 ± 17.3	0.28
Methionine	67.5 ± 7.2	58.7 ± 6.8	66.0 ± 6.3	64.2 ± 5.1	0.78
Tyrosine	199.3 ± 16.6	151.7 ± 17.4	167.1 ± 7.8	180.2 ± 16.3	0.17
Isoleucine	108.7 ± 7.3	93.1 ± 11.2	103.0 ± 6.6	105.7 ± 9.7	0.63
Leucine	142.5 ± 9.8	119.5 ± 15.1	128.8 ± 6.2	144.1 ± 14.9	0.44
Phenylalanine	93.9 ± 7.2	77.6 ± 9.4	75.3 ± 5.7	94.7 ± 10.1	0.22
Tryptophan	58.8 ± 4.4	51.2 ± 4.9	54.2 ± 2.3	58.4 ± 4.8	0.54
7 d					
Asparagine	77.4 ± 4.1	72.0 ± 3.5	75.6 ± 5.3	75.3 ± 3.7	0.83
Aspartic acid	66.2 ± 2.7	68.1 ± 1.4	69.2 ± 2.6	70.7 ± 1.4	0.5
Serine	138.4 ± 11.8	142.2 ± 7.6	147.2 ± 16.4	145.2 ± 6.7	0.95
Alanine	362.5 ± 26.6	329.7 ± 21.1	382.8 ± 23.1	366.3 ± 21.2	0.44
Glycine	687.6 ± 43.7	778.8 ± 40.0	795.1 ± 34.4	742.9 ± 25.9	0.19
Glutamine	370.7 ± 12.0	355.1 ± 17.7	389.0 ± 22.2	374.6 ± 12.3	0.56
Threonine	268.4 ± 12.9	280.9 ± 17.6	285.6 ± 25.6	297.5 ± 10.2	0.71
Cysteine	2.5 ± 0.3	3.0 ± 0.4	2.6 ± 0.5	3.8 ± 0.5	0.18
Glutamic acid	100.6 ± 10.0	101.0 ± 8.7	104.8 ± 6.9	111.8 ± 5.6	0.74
Proline	177.9 ± 10.3	163.0 ± 6.5	173.1 ± 11.8	173.3 ± 8.4	0.72
Lysine	90.0 ± 12.1	56.3 ± 7.3	69.2 ± 17.1	62.3 ± 8.8	0.23
Histidine	25.8 ± 1.8	21.6 ± 1.3	25.7 ± 3.7	23.5 ± 1.4	0.51
Arginine	91.5 ± 9.3	83.2 ± 7.5	92.5 ± 15.2	83.5 ± 7.1	0.87
Valine	223.3 ± 9.9	205.3 ± 11.2	221.5 ± 17.5	231.9 ± 13.3	0.56
Methionine	180.5 ± 29.7	169.0 ± 23.7	156.3 ± 22.6	161.5 ± 18.2	0.9
Tyrosine	136.6 ± 12.9	126.9 ± 8.7	135.0 ± 13.5	150.2 ± 6.6	0.5
Isoleucine	123.2 ± 10.7	105.8 ± 6.8	115.8 ± 8.0	126.9 ± 6.8	0.3
Leucine	123.4 ± 8.5	104.6 ± 6.8	121.1 ± 14.0	118.0 ± 9.3	0.56
Phenylalanine	77.8 ± 5.8	61.1 ± 4.8	70.9 ± 8.2	68.2 ± 4.6	0.29
Tryptophan	45.4 ± 4.1	41.5 ± 3.3	46.9 ± 5.0	49.0 ± 2.5	0.57
14 d					
Asparagine	80.2 ± 5.4	89.1 ± 6.0	83.8 ± 3.4	84.1 ± 8.0	0.77
Aspartic acid	66.5 ± 2.3	73.4 ± 3.3	72.0 ± 1.8	65.0 ± 2.0	0.05
Serine	161.2 ± 13.3	188.3 ± 12.6	173.1 ± 10.8	169.7 ± 15.2	0.53
Alanine	322.3 ± 26.3	342.8 ± 28.7	369.2 ± 20.1	304.0 ± 28.8	0.35
Glycine	1036.6 ± 46.1	1115.9 ± 62.1	1166.7 ± 37.4	1001.7 ± 34.4	0.07
Glutamine	403.1 ± 24.9	431.4 ± 20.5	430.9 ± 18.4	370.9 ± 23.1	0.18
Threonine	315.9 ± 19.1	353.6 ± 18.3	322.2 ± 12.0	328.7 ± 25.9	0.55
Cysteine	3.6 ± 0.5	3.7 ± 0.7	2.5 ± 0.4	4.1 ± 0.5	0.2
Glutamic acid	103.7 ± 9.8	122.3 ± 16.3	116.2 ± 10.1	106.0 ± 7.1	0.62
Proline	196.3 ± 13.4	207.6 ± 12.2	201.6 ± 7.9	185.7 ± 17.6	0.69
Lysine	103.4 ± 12.2	119.6 ± 10.0	105.1 ± 9.6	113.7 ± 16.6	0.77
Histidine	32.7 ± 3.4	34.3 ± 2.4	31.3 ± 2.9	33.0 ± 3.4	0.92
Arginine	88.3 ± 9.4	105.0 ± 8.3	90.5 ± 6.7	92.6 ± 11.4	0.57
Valine	244.9 ± 14.2	263.8 ± 11.3	244.5 ± 10.8	252.1 ± 19.6	0.76
Methionine	157.7 ± 19.6	189.9 ± 27.4	144.8 ± 17.7	156.6 ± 18.9	0.49
Tyrosine	150.6 ± 12.5	162.2 ± 11.5	144.2 ± 7.3	149.0 ± 11.2	0.68
Isoleucine	119.8 ± 9.0	125.3 ± 7.5	122.4 ± 6.6	126.6 ± 11.1	0.95
Leucine	138.9 ± 15.0	146.5 ± 9.4	144.9 ± 10.8	145.3 ± 18.5	0.98
Phenylalanine	71.9 ± 5.4	77.9 ± 6.1	69.1 ± 4.8	71.5 ± 5.7	0.71
Tryptophan	59.0 ± 4.1	64.0 ± 6.4	60.7 ± 3.1	59.1 ± 3.8	0.84
21 d					
Asparagine	121.3 ± 9.6	107.7 ± 7.6	107.6 ± 9.8	122.2 ± 7.1	0.45
Aspartic acid	90.2 ± 2.6	87.1 ± 2.3	89.4 ± 2.9	88.0 ± 2.3	0.83
Serine	246.1 ± 11.8	247.2 ± 20.0	228.4 ± 25.1	278.5 ± 18.4	0.34
Alanine	412.7 ± 36.7	382.8 ± 29.9	421.3 ± 39.3	447.5 ± 17.2	0.56
Glycine	1332.0 ± 97.8	1197.4 ± 34.3	1157.9 ± 34.0	1253.1 ± 53.2	0.22
Glutamine	584.5 ± 31.7	559.9 ± 39.3	571.1 ± 26.2	570.7 ± 16.8	0.95
Threonine	534.6 ± 38.0	506.3 ± 27.2	460.3 ± 38.0	565.8 ± 33.8	0.19
Cysteine	4.2 ± 0.5 ^b^	5.1 ± 0.4 ^ab^	4.8 ± 0.7 ^b^	7.1 ± 0.7 ^a^	0.01
Glutamic acid	117.4 ± 9.3	143.4 ± 10.7	147.5 ± 12.6	135.8 ± 10.6	0.23
Proline	312.4 ± 21.4	296.3 ± 26.0	286.2 ± 29.2	300.3 ± 14.7	0.89
Lysine	181.6 ± 16.4	162.5 ± 13.2	151.3 ± 12.5	196.2 ± 12.9	0.12
Histidine	56.2 ± 3.4	52.3 ± 3.5	51.5 ± 5.9	59.7 ± 4.6	0.55
Arginine	140.2 ± 9.9 ^ab^	141.3 ± 9.9 ^ab^	118.6 ± 8.6 ^b^	164.4 ± 13.8 ^a^	0.04
Valine	428.9 ± 23.2	411.2 ± 30.9	384.4 ± 24.2	436.7 ± 14.3	0.43
Methionine	191.7 ± 27.3	192.5 ± 18.4	187.6 ± 31.4	164.6 ± 12.2	0.81
Tyrosine	208.4 ± 16.5	201.6 ± 14.6	183.8 ± 13.8	206.7 ± 10.1	0.59
Isoleucine	209.5 ± 15.5	191.1 ± 15.8	174.9 ± 11.5	218.5 ± 9.6	0.12
Leucine	253.5 ± 21.9	228.6 ± 21.9	213.7 ± 16.3	271.6 ± 12.2	0.14
Phenylalanine	123.8 ± 9.2 ^a^	102.3 ± 10.1 ^ab^	88.4 ± 7.4 ^b^	121.5 ± 7.4 ^a^	0.02
Tryptophan	99.9 ± 6.6	89.6 ± 7.4	80.7 ± 5.6	96.8 ± 5.4	0.16

All data are expressed as the mean ± SEM; *n* = 9. ^1^ Blood was collected before the provision of feed with MGM-P on the day of weaning. ^a, b^ Mean values within a row with dissimilar superscript letters are significantly different (*p* < 0.05).

**Table 9 animals-14-02337-t009:** Effects of MGM-P supplementation on the organ weight/length of the weaned piglets.

Measurement	NC	LT	HT	PC	*p*-Value
Organ weight/length					
Liver (g)	597.27 ± 86.60	531.93 ± 72.17	514.50 ± 87.61	536.10 ± 45.63	0.87
Pancreas (g)	38.47 ± 5.43	35.67 ± 6.24	35.53 ± 6.16	39.07 ± 5.32	0.96
Spleen (g)	41.47 ± 2.25	47.87 ± 7.60	37.63 ± 2.19	40.67 ± 3.41	0.47
Kidney (g)	135.73 ± 25.50	121.87 ± 22.44	119.10 ± 25.52	128.30 ± 6.33	0.95
Stomach (g)	104.10 ± 18.65	86.87 ± 12.66	87.90 ± 9.40	96.27 ± 10.26	0.78
Small intestine weight (g)	595.97 ± 73.98	603.53 ± 99.37	582.07 ± 36.16	545.03 ± 11.39	0.92
Small intestine length (cm)	1188.00 ± 27.26	1140.17 ± 62.98	1123.00 ± 31.47	1105.00 ± 57.36	0.65
Large intestine weight (g)	202.07 ± 20.78	185.23 ± 28.52	182.50 ± 16.20	177.07 ± 12.16	0.84
Large intestine length (cm)	219.17 ± 10.14	220.00 ± 17.24	215.00 ± 12.58	200.33 ± 10.27	0.69
Relative organ weight/length					
Liver (%)	2.88 ± 0.15	2.65 ± 0.12	2.49 ± 0.06	2.59 ± 0.10	0.18
Pancreas (%)	0.19 ± 0.01	0.18 ± 0.01	0.17 ± 0.02	0.19 ± 0.01	0.78
Spleen (%)	0.21 ± 0.02	0.24 ± 0.00	0.19 ± 0.02	0.20 ± 0.03	0.43
Kidney (%)	0.65 ± 0.03	0.60 ± 0.03	0.57 ± 0.03	0.62 ± 0.03	0.44
Stomach (%)	0.49 ± 0.03	0.43 ± 0.02	0.43 ± 0.03	0.46 ± 0.02	0.33
Small intestine weight (%)	3.10 ± 0.84	3.07 ± 0.48	2.90 ± 0.24	2.66 ± 0.17	0.92
Small intestine length (cm/kg)	59.32 ± 7.93	58.61 ± 7.72	56.61 ± 7.05	53.58 ± 1.60	0.93
Large intestine weight (%)	0.99 ± 0.08	0.92 ± 0.02	0.91 ± 0.08	0.86 ± 0.01	0.50
Large intestine length (cm/kg)	10.86 ± 1.19	11.19 ± 1.10	10.72 ± 0.94	9.86 ± 1.22	0.86

All data are expressed as the mean ± SEM; *n* = 3. There were no statistically significant differences among the four groups based on the results of the one-way analysis of variance.

## Data Availability

The data that support the findings of the present study are available from the corresponding author upon reasonable request.

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
