# Peer review of "Study of the Effects of Condensed Tannin Additives on the Health and Growth Performance of Early-Weaned Piglets"

_animals, 2024, doi:10.3390/ani14162337_

Round 1

Reviewer 1 Report (Previous Reviewer 3)

Comments and Suggestions for Authors

.

Reviewer 2 Report (Previous Reviewer 2)

Comments and Suggestions for Authors

The manuscript has been improved according to the reviewers' suggestions and I therefore express my opinion on the possibility of publishing the manuscripts in the journal Animals.

This manuscript is a resubmission of an earlier submission. The following is a list of the peer review reports and author responses from that submission.

Round 1

Reviewer 1 Report

Comments and Suggestions for Authors

In this study, Min MA et al., aimed to evaluate the effect of a feed additive (0.5% and 1.0% MGM-P) as an alternative to antibiotics for early-weaned piglets. It’s an interesting study, however, some modifications and clarifications are required:

The topic covered is current and of interest to operators in the sector.

Simple Summary

Lines 26 to 28: I suggest you to improve the grammar

Abstract:

Lines 32 to 36: I suggest you to improve the grammar

Lines 40 to 41: I suggest you to improve the grammar

Lines 46 to 47: I suggest you to improve the grammar

I also consider that the abstract should contain some relevant values from the results of the paper and less information related to the author's previous work.

Introduction:

The introduction section of the article is written simply and explains the subject in light of the current literature topics. However, no novelty to the existing state of the field emerges from this study. Please improve this aspect.

Lines 62 to 63: You repeat the information in the simple summary section

Lines 69 to 74: You repeat the information in the abstract section. Please find additional studies to improve this section and to sustain the necessity of your study.

Lines 72 to 74: I suggest you improve the grammar.

Lines 102 to 103: Please verify

It is not well established which is the hypothesis of this study.

Materials and Methods:

The study design is interesting. The experimental part is structured correctly; however, it needs some improvements:

2.1.2. Diet

What do you mean by …….”creep feed”

2.1.3. Animals and Experimental Design

Please add the number of animals in each group.

The animals were housed in groups or individual ?

Based on what criteria did you make your selection with Flavomycin80?

2.2.3. Blood Sampling

Please specify the number of piglets used for blood collection

2.2.1. Diarrhea manifestations

Present the method or provide a reference.

2.2.7. Actual and Relative Weights/Lengths of Organs and Intestines

Can such a few number of pigs detect treatment effects for such variable pig responses?

It is not previously written about the pens and replicates.

Results

The results obtained provide useful information with possible practical applications. They are not clear and concise, however, they need some adjustments:

3.1. Diarrhea

Where are the data??

In the legend of table 2, “n = 3 piglets/group” was not suitable for growth performance. The pen should be used as an experimental unit for growth performance data.

Please add the p value in table 2.

Figure 1.

Why did you choose only the weekly body weight evolution?

Why didn't you present weekly all the data that reflect the zootechnical parameters?

3.3. Blood Hematology Analysis

Lines 179 to 182: I suggest you verify the grammar.

In table 4, please verify the date for the TG (mg/dL) parameter and explain the SEM ??

Please verify the RBC parameter in NC and HT groups; the PLT parameter in NC-HT-PC groups ??

I noticed that due to the small number of animals, there is a very high variability between the results.

Please add the p value in table 3.

3.4. Blood Biochemical Analysis

Please add the p value in table 4.

Please verify the TG parameter in HT and PC groups; The values are almost double without effect ??

Please verify the AMYL parameter in NC and HT groups;

3.5. Plasma free amino acids

Figure 2. (A) and (B)

There is a reason why only the two AA were presented? The statistics revealed differences at the end or when ??

There is a specific reason for selecting only the two AAs that were presented?

Please note that it is difficult to understand the information presented in table 5. I suggest you use landscape.

Please add the p value in table 5 and 6.

Discussion

First of all, the authors discuss the data that are not presented in this paper……PWD….

Lines 232 to 233: I suggest you verify the grammar.

Lines 234 to 236: I suggest you verify the grammar. Also, it needs some improvements or rewrites clearly and completely.

Lines 227, and 251; 260, and 272: I suggest you give other references.

The discussions lack clarity and coherence in expression, making them challenging to understand. The expression and English are very poor. The authors have very few references in discussions and comparisons with data from the literature. The major considerations are necessary.

Conclusions

The conclusions section should be supplemented with more information, limitations of the study, and future perspectives. The conclusion is not supported by the results, PWD is not in this paper.

The references section contains up-to-date articles.

Comments on the Quality of English Language

Extensive editing of English language required

Author Response

Thank you very much.

I would like to respond to your questions carefully. The line numbers changed during the revision process, so all revised parts used blue color.

Simple Summary

Lines 26 to 28: I suggest you to improve the grammar

I will have my manuscript English reviewed by the editing service that animals` recommended to me. 

Abstract:

Lines 32 to 36: I suggest you to improve the grammar

I will have my manuscript English reviewed by the editing service that animals` recommended to me.

Lines 40 to 41: I suggest you to improve the grammar

I will have my manuscript English reviewed by the editing service that animals` recommended to me.

Lines 46 to 47: I suggest you to improve the grammar

I will have my manuscript English reviewed by the editing service that animals` recommended to me.

I also consider that the abstract should contain some relevant values from the results of the paper and less information related to the author's previous work.

Thanks, I revised abstract according to you indicates.

Introduction:

The introduction section of the article is written simply and explains the subject in light of the current literature topics. However, no novelty to the existing state of the field emerges from this study. Please improve this aspect.

Thanks、because our previously published paper (Animals 2021, 11, 3316, doi:10.3390/ani11113316.) described the novelty and emerging of the field so we omitted this description in the present manuscript.

Lines 62 to 63: You repeat the information in the simple summary section

Thanks, it revised.

Lines 69 to 74: You repeat the information in the abstract section. Please find additional studies to improve this section and to sustain the necessity of your study.

Deleted the repeated information in the abstract.

Lines 72 to 74: I suggest you improve the grammar.

I will have my manuscript English reviewed by the editing service that animals` recommended to me.

Lines 102 to 103: Please verify

It is not well established which is the hypothesis of this study.

Thanks, it was revised.

Materials and Methods:

The study design is interesting. The experimental part is structured correctly; however, it needs some improvements:

Thanks, it was revised.

2.1.2. Diet

What do you mean by …….”creep feed”

Thanks, in the present study the creep feed means fed feed before weaned.

2.1.3. Animals and Experimental Design

Please add the number of animals in each group. Thanks, it was revised.

The animals were housed in groups or individual ? Thanks, it was revised.

Based on what criteria did you make your selection with Flavomycin80?

Flavomycin is one of the most using antibiotics in this area and which was the feed company recommended to us. The present results suggest that current feeding conditions might no more need addition of antimicrobial except improve growth performance.

2.2.3. Blood Sampling

Please specify the number of piglets used for blood collection

Thanks, it was revised.

2.2.1. Diarrhea manifestations

Present the method or provide a reference.

How to define the diarrhea is one afflicted problem for us for a long period. According to our knowledge of the pigs` study, the present study authors used our own definition of diarrhea. The diarrhea was defined by sloppy feces having more than 80% moisture that was found on two or more consecutive days. Feces were observed twice daily at 9:00 am and 3:00 pm.

2.2.7. Actual and Relative Weights/Lengths of Organs and Intestines

Can such a few number of pigs detect treatment effects for such variable pig responses?

It is not previously written about the pens and replicates.

Thanks, totally 3 piglets in each treatment group and 12 piglets for all.

Results

The results obtained provide useful information with possible practical applications. They are not clear and concise, however, they need some adjustments:

3.1. Diarrhea

Where are the data??

Thanks, in the manuscript described that no PWD has been confirmed by piglets in all of treatments, including LT, HT, and PC groups and even in the control group respectively.

I am so sorry; I wander is it should be showed by table? Because of all results are the zero.

In the legend of table 2, “n = 3 piglets/group” was not suitable for growth performance. The pen should be used as an experimental unit for growth performance data.

Thanks, the trial had 9 piglets in each group and housed in 3 pens (3 piglets in each pen), so both body weight and ADG were calculated based on individual piglets as a unit (n=9), however feed intake and feed efficiency were calculated based on pens as a unit (n=3). When calculating on a pen basis, although n is only 3, the results obtained for each pen are the average of 3 piglets, and therefore actually account for 9 piglets.

Please add the p value in table 2.

Thanks, the p-value was added.

Figure 1.

Why did you choose only the weekly body weight evolution?

Why didn't you present weekly all the data that reflect the zootechnical parameters?

Thanks, I consider the weight data to be the most important and have listed only the weight data to avoid having too much data that would appear to be too cumbersome. Ultimately, we decided to have added all the performance parameters for each week to the graphs to ensure the completeness of the data.

3.3. Blood Hematology Analysis

Lines 179 to 182: I suggest you verify the grammar.

I will have my manuscript English reviewed by the editing service that animals` recommended to me.

In table 4, please verify the date for the TG (mg/dL) parameter and explain the SEM ?

Please verify the RBC parameter in NC and HT groups; the PLT parameter in NC-HT-PC groups ??

I noticed that due to the small number of animals, there is a very high variability between the results.

Thanks, I checked the raw data of TG, RBC and PLT and ensured that no errors have occurred. Blood was collected from all animals (9 per group, 36 total). The normal ranges of the parameters in the blood are larger, resulting in a larger SEM.

Please add the p value in table 3.

Thanks, the p-value was added.

3.4. Blood Biochemical Analysis

Please add the p value in table 4.

Thanks, the p-value was added.

Please verify the TG parameter in HT and PC groups; The values are almost double without effect ??

Thanks, we have examined the raw data for TG, which is indeed affected by treatments and the results are significantly different.We have discussed this and please review lines 344-346.

Please verify the AMYL parameter in NC and HT groups;

Thanks, we have checked the raw data for AMYL and did not find any errors. There is no further discussion as no significant differences were found.

3.5. Plasma free amino acids

Figure 2. (A) and (B)

There is a reason why only the two AA were presented? The statistics revealed differences at the end or when ??

There is a specific reason for selecting only the two AAs that were presented?

Thanks, only three blood free amino acid concentrations showed significant differences in the last week of the experiment. To allow the reader to quickly grasp the effect of tannin addition on blood free amino acids, we have plotted the concentration changes of two of the more important amino acids. To avoid the same doubt again, we finally decided to graph the concentration changes of all three amino acids that showed significant differences.

Please note that it is difficult to understand the information presented in table 5. I suggest you use landscape.

Thanks, we have adjusted the layout of Table 5 and it now looks more accessible than before.

Please add the p value in table 5 and 6.

Thanks, the p-value was added.

Discussion

First of all, the authors discuss the data that are not presented in this paper……PWD….

Thanks, since diarrhea was not observed in all treatments, thus, the we discussed the possible reason why there is no PWD has happened in piglets in all of treatments.

Lines 232 to 233: I suggest you verify the grammar.

I will have my manuscript English reviewed by the editing service that animals` recommended to me.

Lines 234 to 236: I suggest you verify the grammar. Also, it needs some improvements or rewrites clearly and completely.

I will have my manuscript English reviewed by the editing service that animals` recommended to me.

Lines 227, and 251; 260, and 272: I suggest you give other references.

The discussions lack clarity and coherence in expression, making them challenging to understand. The expression and English are very poor. The authors have very few references in discussions and comparisons with data from the literature. The major considerations are necessary.

Thanks, because of less study have done about use MGM-P and other condensed tannin for the monogastric animals, so added only another one reference in the discussion. And I will have my manuscript English reviewed by the editing service that animals` recommended to me.

Conclusions

The conclusions section should be supplemented with more information, limitations of the study, and future perspectives. The conclusion is not supported by the results, PWD is not in this paper.

Thanks, I revised this section.

The references section contains up-to-date articles.

Thank you for your kindness.

Reviewer 2 Report

Comments and Suggestions for Authors

The abbreviation - MGM-P should be explained in the summary

A large part of the manuscript is devoted to parameters indicating the health of animals, so I believe that such information should be included in the title of the work.

In chapter 2.1.3. missing information:

- how were the animals kept in the group?

- how many repetitions were there per group?

- how was feed intake tested? - was it individual or group consumption? - if group, for how many animals?

Were the animals fasting before blood collection?

The authors did not provide information on how the animals were fed.

Line 141 - Did I understand correctly that the organ analysis was based on one individual? - if I think so, it is not reliable, such tests should be performed on at least 6 individuals. Otherwise, the test should be repeated.

The authors did not provide the component and chemical composition of feed mixtures - this should be supplemented. In particular, they refer to this information in lines 234-236.

On what basis do the authors claim that the feed "...intestinal mucosa villus." since no such research has been carried out Line 238

Unfortunately, I don't understand any of this - Did the feed used contain "... like probiotics, several other kinds of herbal extract, and little bit high level of zinc sulfate.." did it have it? - it's a bit complicated - it needs to be sorted out (Line 240).

In the discussion, the authors refer to the effect of tannins in ruminants. I believe that this is not an appropriate comparison because polygastric animals, due to the presence of microorganisms, deal with antinutritional substances in a completely different way than monogastric animals - the discussion should be verified.

I believe that the work should be supplemented with the information included in the comments.

First of all, the authors should reword the discussion focusing mainly on monogastric animals.

Author Response

Thak you very much.

I would like to respond to your questions carefully. The line numbers changed during the revision process, so all revised parts used blue color

The abbreviation - MGM-P should be explained in the summary

Thanks, I have added in simple summary section.

MGM is a commercial brand, it came from the italian word for feed: ManGiMe, P represent pig.

A large part of the manuscript is devoted to parameters indicating the health of animals, so I believe that such information should be included in the title of the work.

Thanks, the title has been changed to " Studies on the effects of condensed tannin additives on the health and growth performance of early weaned piglets ".

In chapter 2.1.3. missing information:

- how were the animals kept in the group?

- how many repetitions were there per group?

- how was feed intake tested? - was it individual or group consumption? - if group, for how many animals?

Thanks, it was revised.

Were the animals fasting before blood collection?

Thanks, We did not fast the piglets before blood collection.

The authors did not provide information on how the animals were fed.

Thanks, I added information detail.

Line 141 - Did I understand correctly that the organ analysis was based on one individual? - if I think so, it is not reliable, such tests should be performed on at least 6 individuals. Otherwise, the test should be repeated.

Thanks, I am so sorry for not presenting this clearly. The 9 piglets in each group were divided into 3 pens for feeding, and at the end of the experiment one piglet closest to the average weight was selected from each pen for dissection, thus 3 piglets were dissected from each group, which was a small number but allowed for statistical analysis.

The authors did not provide the component and chemical composition of feed mixtures - this should be supplemented. In particular, they refer to this information in lines 234-236.

Thanks, the current study was an extension of a previous study and therefore used the same feeds as the previous study (Animals 2021, 11, 3316, doi:10.3390/ani11113316.). The mixture composition and chemical composition have been described in previous study, which is cited in the description to avoid repetition.

On what basis do the authors claim that the feed "...intestinal mucosa villus." since no such research has been carried out Line 238

Thanks, has removed.

Unfortunately, I don't understand any of this - Did the feed used contain "... like probiotics, several other kinds of herbal extract, and little bit high level of zinc sulfate.." did it have it? - it's a bit complicated - it needs to be sorted out (Line 240).

Thanks, we used the commercial feed brand for weaning stage, that composition contained probiotics, several other kinds of herbal extract, and little bit high level of zinc sulfate. This is the reason we plan do next experiment design feed formulation by ourself.

In the discussion, the authors refer to the effect of tannins in ruminants. I believe that this is not an appropriate comparison because polygastric animals, due to the presence of microorganisms, deal with antinutritional substances in a completely different way than monogastric animals - the discussion should be verified.

Thanks, it have removed.

I believe that the work should be supplemented with the information included in the comments.

First of all, the authors should reword the discussion focusing mainly on monogastric animals.

 Thanks, for your kind indications.  

Reviewer 3 Report

Comments and Suggestions for Authors

The objective of the study was to evaluate the adventages of addition condensed tanins to feed for young piglets. Growth performance, frequency of diarhea incidence, blood hematology and biochemistry, plasma amino acids concentration and weights and lengths of organs and gut were used as parameters.  That manuscript is in scope of journal, however it needs many changes and explanation before acceptation for publication. Below you can find my suggestion for consideration:

1.     I suggest to change title to be informative and atractive. Title should not contain abreviation. I propose to use „ condensed tanin instead of „vegetable extraction quebracho tannin product MGM-P”. It was use only two levels than „different additive level” is not justified. I suggest also add other determined parameters.

2.     Simple Summary and Abstract are not informative. They should be focused on the current study, but not on previous. Please, correct them.

3.     Please, explain why only two levels have been used and described the reasen of choosing.

4.     Hypothesis is missing. Please, add it.

5.     Please, add p-values to tables 2-6. All abbreviations used in tables should be explained below table.

6.     Mean of which results is presented in table 1. Initial BW?

7.     From statistical point of view the number of observation (n=3) is too small.

8.     Please, add information on experiment, how many piglets were kept in one pen, how many pens were used per group

9.     Table 5 contains to omany data and it is difficult to follow it

10.  Comparision of tannin effect between animal species is not good idea.

11.  Please, correct conclusion. Tannin addition did not affect any determined parameter and cannot be proposed as alternative to antibiotic additives for weaned piglets.

Author Response

Thanks, for your kind indications.

I would like to respond to your questions carefully. The line numbers changed during the revision process, so all revised parts used blue color.

  1. I suggest to change title to be informative and atractive. Title should not contain abreviation. I propose to use „ condensed tanin instead of „vegetable extraction quebracho tannin product MGM-P”. It was use only two levels than „different additive level” is not justified. I suggest also add other determined parameters.

Thanks, I have changed the title of the paper to " Studies on the effects of condensed tannin additives on the health and growth performance of early weaned piglets ". And revised.

  1. Simple Summary and Abstract are not informative. They should be focused on the current study, but not on previous. Please, correct them.

Thanks, because the present study follows on from our previous work, a brief description of the previous work can further illustrate the significance of the present study. We have taken your suggestion to present information related to previous work as succinctly as possible in the Simple Summary only, while removing relevant content entirely from the abstract to focus on the content and presentation of results of the present study.

  1. Please, explain why only two levels have been used and described the reasen of choosing.

Thanks, currently there is limited research on the use of quebracho tannins in piglet diets, and we have only been able to try them gradually at 0.1%, 0.2%, and 0.3%, and have found that 0.3% is more effective. Therefore, we would like to try higher dosages to enhance the benefits, but the antinutritional effects of tannins may also concerned. To avoid antinutritional effects, we set the maximum value at 1% because one study showed that 1% condensed tannins do not have antinutritional effects on pigs. We also chose to use half the dose (0.5%), both because the higher dose allowed for a larger range and because it allowed for more possibilities to be discovered in conjunction with our previous research. We eventually found that the 1% dosage already showed antinutritional effects, while the 0.5% dosage was beneficial to piglet growth, so our design is scientific and reasonable.

  1. Hypothesis is missing. Please, add it.

Thanks, it has revised.

  1. Please, add p-values to tables 2-6. All abbreviations used in tables should be explained below table.

Thanks, the p-values added to the Tables. And explained all the abbreviations used in the table below the table.

  1. Mean of which results is presented in table 1. Initial BW?

Thanks, the experiment started immediately after grouping, so it is the average of the initial weight. The procedure was a simulation of a pig farm procedure and therefore was professionally performed to avoid stress.

  1. From statistical point of view the number of observation (n=3) is too small.

Thanks, each group in our study consisted of 9 piglets kept in three pens. parameters with n of 3 were ADFI, FCR, fecal scores, and organ indexes. ADFI, FCR, and fecal scores measurements were made in pens, so that the values obtained were actually the average of the three animals, and despite the n of 3, 9 animals were included. For the organ indexes, we sacrificed only 3 piglets per treatment group, it was minimally number of piglets but possible to statistically differ. In each pen, we chose the piglets that were closest to the average weight to be autopsied, and we made sure that the sexes were the same. In addition, our team is very specialized in dissections and has the ability to avoid errors as much as possible.

  1. Please, add information on experiment, how many piglets were kept in one pen, how many pens were used per group

Thanks, it has revised.

  1. Table 5 contains to many data and it is difficult to follow it

Thanks, I have adjusted the layout of Table 5 to make it easier for you to search for information clearly. In addition, I have plotted the data with significant differences in Figure 2 to make it easier for the reader.

  1. Comparision of tannin effect between animal species is not good idea.

Thanks, I have removed most of the discussion related to ruminants and focused the discussion on monogastrics.

  1. Please, correct conclusion. Tannin addition did not affect any determined parameter and cannot be proposed as alternative to antibiotic additives for weaned piglets.

Thanks, for your kind indications, hopefully this correction could satisfactory to your request.

Round 2

Reviewer 1 Report

Comments and Suggestions for Authors

Dear Editor,

The author's response was convincing, and I have no further comments.

Author Response

Thanks again, for kind consideration.

Reviewer 3 Report

Comments and Suggestions for Authors

Dear Authors,

1. Three observations per groups is not acceptable in scietific papers.

2. Table 1 was not corrected.

3. Tables 2-6 do not contain p-value.

4. Now Table 5 is the worse.

Author Response

Thanks again, and I answered your question as below.

  1. Three observations per groups is not acceptable in scietific papers.

Thanks, in the present study every treatment have the 9 observations, except average daily feed intake (ADFI) and feed conversion rate (FCR) were calculated which was calculated by pen (3 pens/group).

  1. Table 1 was not corrected.

Thanks, I corrected Table 1. This time it should be easy understanding, 9 piglets per treatment and fed in 3 pens. All the measurement we recorded one by one except feed intake (recorded by pen unit).

  1. Tables 2-6 do not contain p-value.

Thanks, the p-values have added.

  1. Now Table 5 is the worse.

Thanks, we revised the table 5 again and hopefully this version satisfy your required.
